# Structural basis of Nipah virus RNA synthesis

Fernanda A. Sala[1,2], Katja Ditter[1,2], Olexandr Dybkov[3], Henning Urlaub ®[3,4,5] & Hauke S. Hillen ®[1,2,5,6] ✉

Nipah virus (NiV) is a non-segmented negative-strand RNA virus (nsNSV) with high pandemic potential, as it frequently causes zoonotic outbreaks and can be transmitted from human to human. Its RNA-dependent RNA polymerase (RdRp) complex, consisting of the L and P proteins, carries out viral genome replication and transcription and is therefore an attractive drug target. Here, we report cryo-EM structures of the NiV polymerase complex in the apo and in an early elongation state with RNA and incoming substrate bound. The structure of the apo enzyme reveals the architecture of the NiV L-P complex, which shows a high degree of similarity to other nsNSV polymerase complexes. The structure of the RNA-bound NiV L-P complex shows how the enzyme interacts with template and product RNA during early RNA synthesis and how nucleoside triphosphates are bound in the active site. Comparisons show that RNA binding leads to rearrangements of key elements in the RdRp core and to ordering of the flexible C-terminal domains of NiV L required for RNA capping. Taken together, these results reveal the first structural snapshots of an actively elongating nsNSV L-P complex and provide insights into the mechanisms of genome replication and transcription by NiV and related viruses.

Nipah virus (NiV) is an emerging zoonotic pathogen of the *Paramyxoviridae* family within the order *Mononegavirales*, which encompasses nonsegmented, negative-strand RNA viruses (nsNSVs). Members of this order also include other important human pathogens such as respiratory syncytial virus (RSV), vesicular stomatitis virus (VSV), mumps virus (MUV), rabies virus (RABV), and Ebola virus (EBOV)[1]. Since its initial identification in the 1990s in Malaysia and Singapore[2], NiV has caused recurrent outbreaks, the most recent in Bangladesh and India. The virus represents a major public health concern due to its high mortality rates (up to 80%) and human-to-human transmission[3]. Currently, no vaccines or targeted antiviral treatments to combat Nipah virus infection and disease are available. As a result, the World Health Organization (WHO) has classified the Nipah virus as a priority for epidemic preparedness as part of its Blueprint for Action to Prevent Epidemics[4].

Central to the life cycle of NiV is its RNA-dependent RNA polymerase (RdRp) complex. As in other nsNSVs, it is comprised of the L protein, a large multi-functional polypeptide that contains all required enzymatic activities, and its cofactor, the P protein (called VP35 in *Filoviridae*)[5]. This machinery carries out both genome replication as well as transcription of individual genes. In both cases, the L-P complex initiates RNA synthesis de novo at a leader sequence (*Le*) at the 3' end of the (−) sense genome. For transcription, the L-P complex then undergoes re-initiation at the start of each of the six genes and produces capped and poly-adenylated mRNAs. By contrast, during replication, the L-P complex produces a full-length (+) sense antigenome, which in turn serves as the template for the synthesis of a (−) sense genome copy[6]. Due to its central role for both transcription and replication of the Nipah genome, the L-P complex represents an attractive target for the development of antiviral compounds.

[1]Department of Cellular Biochemistry, University Medical Center Göttingen, Göttingen, Germany. [2]Research Group Structure and Function of Molecular Machines, Max Planck Institute for Multidisciplinary Sciences, Göttingen, Germany. [3]Bioanalytical Mass Spectrometry Group, Max Planck Institute for Multidisciplinary Sciences, Göttingen, Germany. [4]Bioanalytics Group, Institute for Clinical Chemistry, University Medical Center Göttingen, Göttingen, Germany. [5]Cluster of Excellence "Multiscale Bioimaging: from Molecular Machines to Networks of Excitable Cells" (MBExC), University of Göttingen, Göttingen, Germany. [6]Göttingen Center for Molecular Biosciences (GZMB), Research Group Structure and Function of Molecular Machines, University of Göttingen, Göttingen, Germany. ✉e-mail: hauke.hillen@med.uni-goettingen.de

The L protein is comprised of three enzymatic and two structural domains: the RNA-dependent RNA polymerase (RdRp) domain, responsible for catalyzing RNA synthesis, the GDP poly-ribonucleotidyltransferase (PRNTase or CAP) domain involved in cap addition, the connector domain (CD), the methyltransferase (MTase) domain crucial for cap methylation, and the C-terminal domain (CTD)[7]. The P protein consists of an N-terminal domain (NTD), a central oligomerization domain (OD), and a C-terminal X domain (XD).

Over the past years, structures of L-P complexes from several members of the *Paramyxoviridae* family as well as from other nsNSVs have been determined in the apo state without nucleic acids bound[8–15]. In addition, structures of EBOV and RSV L-P complexes in complex with leader RNA have recently been obtained[16,17]. These structures have shed light on the overall architecture of nsNSV L-P complexes and provided first insights into how they interact with the template RNA during initiation. However, to date, no structures of nsNSV L-P complexes in their active state with template and product RNA bound have been reported. As a consequence, the molecular mechanisms underlying nsNSV genome replication and transcription and how these different functions are regulated remain poorly understood.

Here, we present single-particle cryo-electron microscopy (cryo-EM) structures of the NiV L-P complex in its apo state as well as in an early elongating state, with template RNA, product and incoming nucleotide triphosphate (NTP) bound. These structures provide molecular insights into Nipah virus RNA synthesis and represent the first structure of an nsNSV polymerase in the elongation state. Comparison between the two structures reveals key functional elements that undergo conformational rearrangements upon RNA binding and early elongation and show large-scale structural reorganization of the L-P complex, including ordering of the otherwise flexible C-terminal domains. Taken together, these results provide a framework for understanding and targeting Nipah virus replication and transcription and represent an important milestone in our understanding of the reaction cycle of nsNSV polymerases in general.

## Results

### Structure of the apo NiV L-P complex

To obtain structural insights on the NiV L-P complex, we co-expressed codon-optimized NiV L and P proteins in insect cells. Purification via an affinity tag on the NiV L protein led to co-purification of NiV P protein, and we further purified the complex to homogeneity by heparin and size exclusion chromatography and confirmed its identity using mass-spectrometry (Supplementary Fig. 1a, b, "Methods"). Incubation of the complex with a synthetic single-stranded RNA comprising the NiV 3′ leader sequence (*Le*) and substrate NTPs led to the formation of RNA products, showing that the purified NiV L-P complex is active (Supplementary Fig. 1c). We then analyzed the purified L-P complex by single-particle cryo-EM, which led to a reconstruction at an overall resolution of 2.6 Å (Supplementary Fig. 2 and Supplementary Table 1). This allowed us to build and refine an atomic model of the apo NiV L-P complex with excellent stereochemistry (Fig. 1 and Supplementary Table 1).

The overall structure of the NiV L-P complex resembles that previously observed for other nsNSV L-P complexes[5,6]. It is comprised of one copy of NiV L and four copies of NiV P (Fig. 1a,b,c), a stochiometry that was previously observed for L-P complexes of other paramyxoviruses[13–15,18], rhabdoviruses[8,19], pneumoviruses[11,12] and filoviruses[10]. We observe clear density for the RdRp and PRNTase domains of NiV L (residues 1-1452), which together adopt a globular fold with a large cavity at its center. By contrast, the C-terminal CD, MTase, and CTD of L are not visible, indicating that these domains are conformationally flexible in the apo state of the enzyme (Fig. 1a). This is consistent with structural studies on some other nsNSV L-P complexes, which also did not observe density for the corresponding parts of L[10–12,20]. For the NiV P protein, we observe density for the OD

domains (residues 470–578) of all four copies in the complex, which assemble into a rod-like structure that protrudes from the RdRp core. For one of the NiV P protomers (P1), we observe additional density for a linker and the C-terminal X domain (XD) (Supplementary Fig. 2f), which binds at the side of the NiV L RdRp domain. The remaining parts of NiV P are not visible in our maps, indicating conformational flexibility. This is consistent with several apo NiV L-P structures that were reported during the preparation of this manuscript[21–25] as well as with structures of related nsNSV L-P complexes, in which large parts of P were also not visible[8,11–15,18,19].

The RdRp domain of NiV L adopts the canonical, highly conserved right-hand fold characteristic of single-subunit RNA polymerases, composed of thumb, fingers, and palm subdomains (Fig. 1d). The active site is located within a large cavity at the center of the RdRp domain and contains seven structural motifs (A-G) conserved across viral RdRps (Supplementary Fig. 3a)[26]. The catalytic GDNE motif (res. 831–834) resides within motif C, which is part of the palm subdomain (Fig. 1d and Supplementary Fig. 3a). In addition, the RdRp domain contains two conserved elements, the supporting helix (res. 588–600) and the supporting loop (res. 579–587), which have been observed to adopt different conformations associated with distinct functional states in other nsNSV polymerase complexes[17,27]. While the supporting loop is partially ordered in the apo NiV L-P complex, the supporting helix appears to be flexible (Supplementary Fig. 3b). Compared to other nsNSV RdRps, NiV L contains a large sequence insertion of unknown function in its palm subdomain (res. 603–711) (Supplementary Fig. 4) for which we do not observe density, indicating that it is conformationally flexible. The active site is accessible to the solvent through four channels, which presumably serve as NTP entry, template entry, template exit and product exit sites, respectively (Supplementary Fig. 3c). The PRNTase domain required for RNA capping resides on top of the RdRp domain and forms a lid over the active site cavity (Fig. 1c). It contains two zinc-binding motifs (Supplementary Fig. 4), which have been proposed to serve a primarily structural role in the related VSV L-P complex[9]. It further contains two conserved structural elements: the priming loop (res. 1266–1290) and the intrusion loop (res. 1337–1362). The priming loop may be involved in initiation by interacting with the first incoming nucleotides[28,29]. The intrusion loop contains a catalytic HR motif, which forms a covalent bond with the 5′ end of the nascent RNA during capping[30,31]. Both elements have been observed to be either disordered or to adopt different conformations in apo structures of other viral polymerase complexes (Supplementary Fig. 3d)[8–10,13,14,18,19,27]. In the apo NiV L-P complex, both the priming and intrusion loop display only weak density, indicating that they are largely flexible. Thus, neither of these elements appear to be ordered in the absence of nucleic acids.

The four protomers of NiV P form a characteristic stalk that extends from the polymerase core and is formed by coiled-coil interactions between their OD domains (Fig. 1b, c), reminiscent of the structure of P/VP35 in other nsNSV polymerase complexes[6]. A unique feature of the NiV P is a mushroom-like tip at the distal end of the stalk. Although this region is not well resolved in our reconstruction, we could model it with the help of a previous crystal structure of the NiV P OD domain tetramer[32]. This shows that the cap is formed by res. 478–506 of NiV P, which form helices that fold back onto the coiled-coil OD bundle (Fig. 1c and Supplementary Fig. 3e). The base of the P-stalk is anchored to the polymerase core mainly through interactions between the RdRp domain of NiV L and P3 and P4, respectively (Fig. 1e). The XD domain of P1 folds into a three-helix bundle that clamps onto the side of the NiV L RdRp domain, near the substrate NTP entry channel (Fig. 1e). This is consistent with previous structures of EBOV[10] and hPIV5[14] apo L-P complexes, in which the XD domains of VP35 or P were observed to bind in similar positions.

Overall, the structure of the apo NiV L-P complex shows that it adopts a highly conserved architecture resembling that of other nsNSV

L-P complexes and that its C-terminal domains as well as key functional elements in the RdRp core are mobile in the absence of nucleic acids.

## Structure of an actively elongating NiV L-P complex

In order to obtain more detailed insights into the structural basis of Nipah virus RNA synthesis, we next aimed to capture a structural snapshot of NiV L-P in complex with template and product RNA. To achieve this, we incubated the recombinant NiV L-P complex with a synthetic RNA comprising the NiV *Le* sequence in the presence of ATP, CTP, UTP, and the β,γ non-hydrolyzable GTP analog Guanosine-5′-[(β,γ)-imido]triphosphate (GMPPNP). Single-particle cryo-EM analysis of this sample revealed two major particle populations (Supplementary Fig. 5). The first corresponds to the above-described apo NiV L-P complex, with no RNA visible in the active site and disordered OD, MTase, and CTD domains. The second shows clear extra density for a duplex RNA in the active site, thus representing an RNA-bound NiV L-P complex. Refinement of this particle population led to a consensus reconstruction at an overall resolution of 2.8 Å. Multibody refinement in RELION led to improved local maps for the RdRp core, the P-stalk, and the C-terminal domains, which enabled us to build a molecular model of the RNA-bound NiV L-P complex (Supplementary Table 1).

The core of the RNA-bound NiV L-P complex adopts an identical overall structure as the apo enzyme (r.m.s.d 0.971 Å) (Fig. 2a–c and Supplementary Fig. 6a). Compared to the apo NiV L-P complex, the reconstruction showed improved density for several regions of NiV P. First, Multibody refinement with a mask around the P-stalk led to an improved map for the mushroom-like tip of the stalk (Supplementary Fig. 6b). Second, we were able to model a segment C-terminal of the OD domain of P3, which meanders along the side of the NiV L RdRp domain (Supplementary Fig. 6c). The P stalk adopts a slightly different orientation with regards to NiV L than in the apo structure, suggesting that it is conformationally dynamic (Supplementary Fig. 6d). Most strikingly, in contrast to the apo NiV L-P complex structure, we observe clear density for the previously disordered flexible domains of NiV L comprising the OD, MTase, and CTD (Fig. 2a–c). Thus, these domains appear to become ordered upon RNA binding and synthesis. In the active site, characterized by the conserved right-hand fold domain containing the catalytic motifs (A-G), we observe density for a 9-base pair RNA duplex composed of the NiV *Le* template as well as product RNA (Fig. 2d, e and Supplementary Fig. 5f). The resolution is sufficient to unambiguously discriminate between purine and pyrimidine bases, revealing that the NiV polymerase complex has synthesized RNA up to the first occurrence of a cytidine base in the template strand (C10, numbering from 3′ to 5′) in situ. In addition, we observe density for three additional downstream nucleotides of the template RNA (C10-C12). The 3′ end of the nascent RNA is positioned in the −1 site and the +1 site is occupied by an incoming NTP, which base pairs with the templating cytidine base (C10) and which we modeled as GMPPNP (Fig. 2d). We observe clear density for the sugar, base and for all three phosphates of GMPPNP, but no connection to the 3′ end of the nascent RNA is visible at reasonable map threshold, suggesting that we captured the incoming substrate prior to phosphodiester bond formation (Fig. 2d). Thus, the structure represents an actively synthesizing NiV L-P complex during early RNA elongation, captured in the post-translocated state with incoming NTP bound. This state has, to our knowledge, not been described for any nsNSV polymerase so far.

## Interactions between the NiV L-P complex and RNA during elongation

The structure of the elongating NiV L-P complex reveals how the enzyme interacts with nucleic acids during early RNA synthesis. The template RNA enters the active site through the template entry channel, which is located at the interface between the RdRp and PRNTase domains (Supplementary Fig. 6e). Comparison between the apo and elongating NiV L-P structures shows that in the RNA-bound state, the

fingers domain undergoes a shift of approximately 4 Å, which may stabilize the entering template RNA (Supplementary Fig. 6a). The template strand follows the same trajectory as observed in the EBOV and RSV initiation complexes[16,17], suggesting that no large rearrangements of the template occur between initiation and the early stages of RNA synthesis (Supplementary Fig. 6f).

The active site accommodates a 9-base pair (bp) RNA duplex, which is stabilized by interactions with residues from the palm, finger, thumb, and PRNTase domain (Fig. 3a, b). The upstream edge of the duplex faces towards a wall formed by res. 1050–1059 and 1117–1125, which likely limits the length of duplex that can be accommodated. The 5′ end of the product RNA faces towards the RNA exit channel, which is formed between the PRNTase and RdRp domains (Supplementary Fig. 6e). In the active center of the enzyme, the templating base C10 is stabilized in the +1 position by stacking against the conserved residue F553 in motif F of the fingers domain (Fig. 3a and Supplementary Fig. 4). The incoming GMPPNP is stabilized through base pairing with the templating C10. In addition, it stacks against R551 from motif F, which protrudes towards the NTP binding site and is also strictly conserved in nsNSV polymerases (Fig. 3a and Supplementary Fig. 4). The triphosphate moiety of the incoming NTP binds in a groove formed by the catalytic loop in motif C (residues 826–837), which contains one of the two conserved catalytic aspartate residues (D832), and motif A, which contains a second conserved catalytic aspartate (D722). Comparison between the apo and RNA-bound states of NiV L-P shows that motif C repositions to accommodate the product RNA (Fig. 3c). Both catalytic aspartates face towards the triphosphate and complex one of the two catalytic Mg ions, for which we observe density in between the aspartates and the triphosphate moiety (Fig. 2d). This is consistent with biochemical data, which show that mutations of D832 impair the polymerization activity of NiV L-P[22,25,33]. On the opposing side, the triphosphate of the incoming NTP is stabilized by interactions with R551. Structural comparison to other viral polymerases for which elongating structures have been reported, such as SARS-CoV-2[34] and Influenza A[35], shows that the architecture of the active site and arrangement of catalytic residues is highly conserved (Supplementary Fig. 6g).

Taken together, the structure of elongating NiV L-P reveals how it interacts with RNA template and product strands as well as the incoming NTP during early RNA synthesis and shows that the architecture of the active site is highly conserved.

## Structure of the CD-MTase-CTD module of the NiV L-P complex

The cryo-EM reconstruction of the elongating NiV L-P complex also reveals the structure of the NiV L CD, MTase and C-terminal domains, which are flexible in the apo state of the enzyme but become ordered in the RNA-bound state (Figs. 2, 4a). The CD features a conserved compact architecture composed mainly of alpha helices (Fig. 4a, Supplementary Fig. 7a). It is connected to the adjacent PRNTase and MTase domains by linkers at both ends of the CD (residues 1453–1469 and 1746–1758). The N-terminal linker is partially ordered (res. 1453–1463) and the C-terminal linker is disordered, suggesting that they may act as flexible hinges between the domains, as previously suggested[14]. The MTase domain of NiV L adopts a characteristic fold comprised of an eight-stranded β-sheet flanked by helices, which is structurally conserved across nsNSV L-P complexes[13–15,18,19] (Supplementary Fig. 7b). It serves as a dual-function enzymatic domain during co-transcriptional capping by methylating the GTP cap first at the 2′-O position and then at the N7 position[30,36,37]. The binding site for the methyl donor S-adenosylmethionine (SAM) contains a conserved GxGxG motif (residues 1843–1847), and a second set of conserved charged residues K-D-K-E (K1821, D1940, K1976, E2013) forms the catalytic tetrad required for methyl group addition (Fig. 4b and Supplementary Fig. 4)[37–39]. The CTD domain, while not strongly conserved at the sequence level, is structurally homologous to those found in

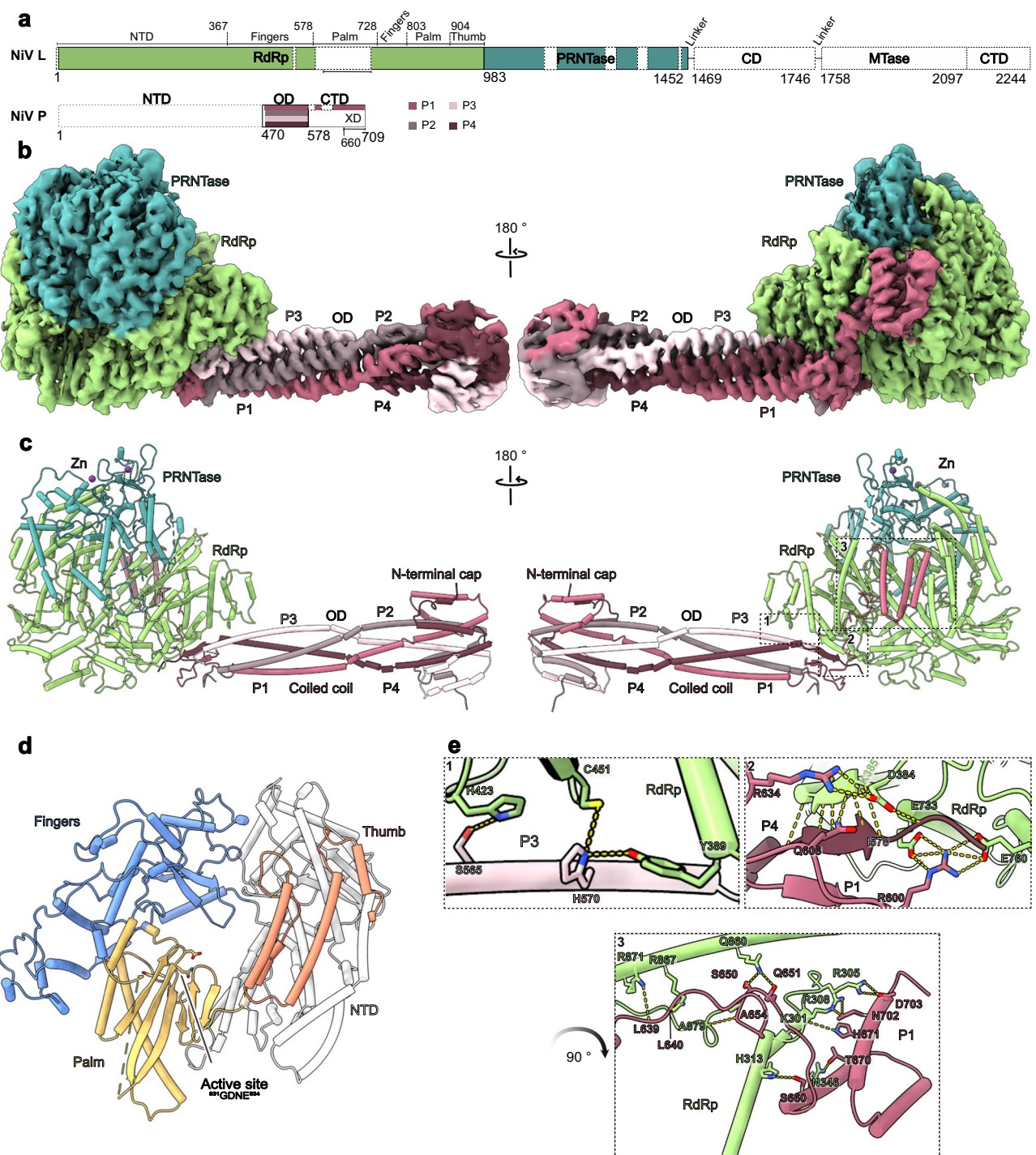

**Fig. 1 | Structure of the apo NiV L-P complex. a** Schematic domain representation of NiV L and P protein. The RdRp domain of NiV L is shown in green and the PRNTase domain in dark cyan. The four P proteins are shown in varying shades of pink. Unmodelled regions are indicated as white dashed boxes. Numbers indicate amino acid positions at domain boundaries. This color scheme is consistently used throughout. **b** Cryo-EM density map of the apo NiV L-P complex (Local resolution filtered map). **c** Cartoon representation of the apo NiV L-P complex with Zn²⁺ ions shown as purple spheres. **d** Structure of the NiV RdRp domain colored by subdomains. The fingers domain is blue, the thumb domain is salmon, the palm domain is yellow gold, and the NTD is gray. The active site, represented by the GDNE motif (residues 831–834), is shown as sticks. **e** Close-up view of interactions between the NiV P and L proteins. The interactions are categorized into three regions (1–3) indicated in (**c**). Interacting side chains are shown as sticks, with hydrogen bonds highlighted by yellow dashed lines. (1) Interaction between P3 an L RdRp. (2) Interactions among P1, P4, and L proteins. (3) Interaction between P1 XD and L RdRp. The arrow indicates rotation of the view showing region 3 relative to 1 and 2.

other nsNSV polymerases (Supplementary Figs. 4, 7b). Its most C-terminal helix stacks against the MTase domain, forming a narrow, positively charged RNA-binding groove. It contains a partially conserved KKG motif (K2232, K2236, G2239), which faces towards the active site of the MTase domain and has been implicated in both methylation and capping (Fig. 4b)[37,40]. Thus, the CTD likely plays in important role in capping during NiV transcription, as reported for other nsNSVs[37,40–42].

In previously reported structures of apo nsNSV L-P complexes, the CD, MTase, and CTD were either not visible or observed to adopt

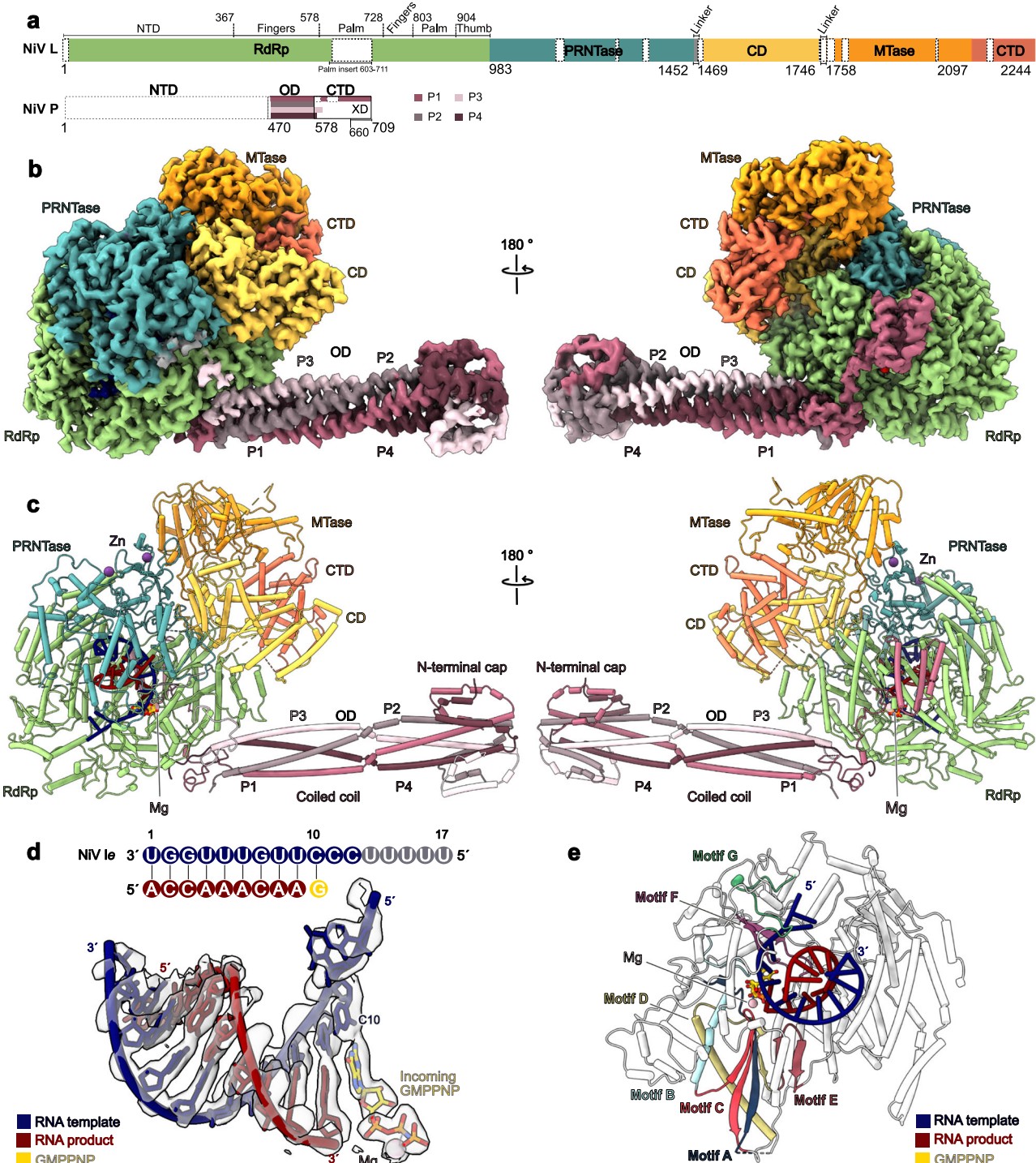

**Fig. 2 | Structure of the actively elongating NiV L-P complex. a** Schematic domain representation of NiV L and P protein. The RdRp domain of NiV L is shown in green, the PRNTase domain in dark cyan, the CD in yellow, the MTase domain in gold yellow, and the CTD in orange. The four P proteins are shown in varying shades of pink. Unmodelled regions are indicated as white dashed boxes. This color scheme is consistently used throughout. **b** Composite cryo-EM density map of the actively elongating NiV L-P complex. **c** Cartoon representation of the actively elongating NiV L-P complex. The template RNA is shown in blue and the product RNA in red. $Zn^{2+}$ and $Mg^{2+}$ are shown as purple and pink spheres, respectively. The color code is used throughout. **d** Schematic and stick representation of the RNA template and product with incoming GMPPNP (yellow) and corresponding cryo-EM density. **e** Structure of the NiV L RdRp domain with RNA bound with conserved catalytic motifs color-coded from A to G. The active site GDNE motif (residues 831–834) and incoming GMPPNP are shown as sticks.

different orientations relative to the RdRp and PRNTase domains[6]. Comparisons to these structures show that the arrangement of the C-terminal parts of NiV L-P in the RNA-bound complex most closely resembles that seen in the hPIV3 L-P and MUV L-P complexes[13,15], while differing from the orientation observed in the PIV5 L-P complex (Supplementary Fig. 8a)[14]. Both the MUV L-P and PIV5 L-P complexes have been proposed to resemble transcribing states. In the case of MUV L-P, this assumption was based on the presence of a continuous cavity from the RdRp active site to the MTase active site, which would suggest a potential path for the nascent RNA during co-transcriptional

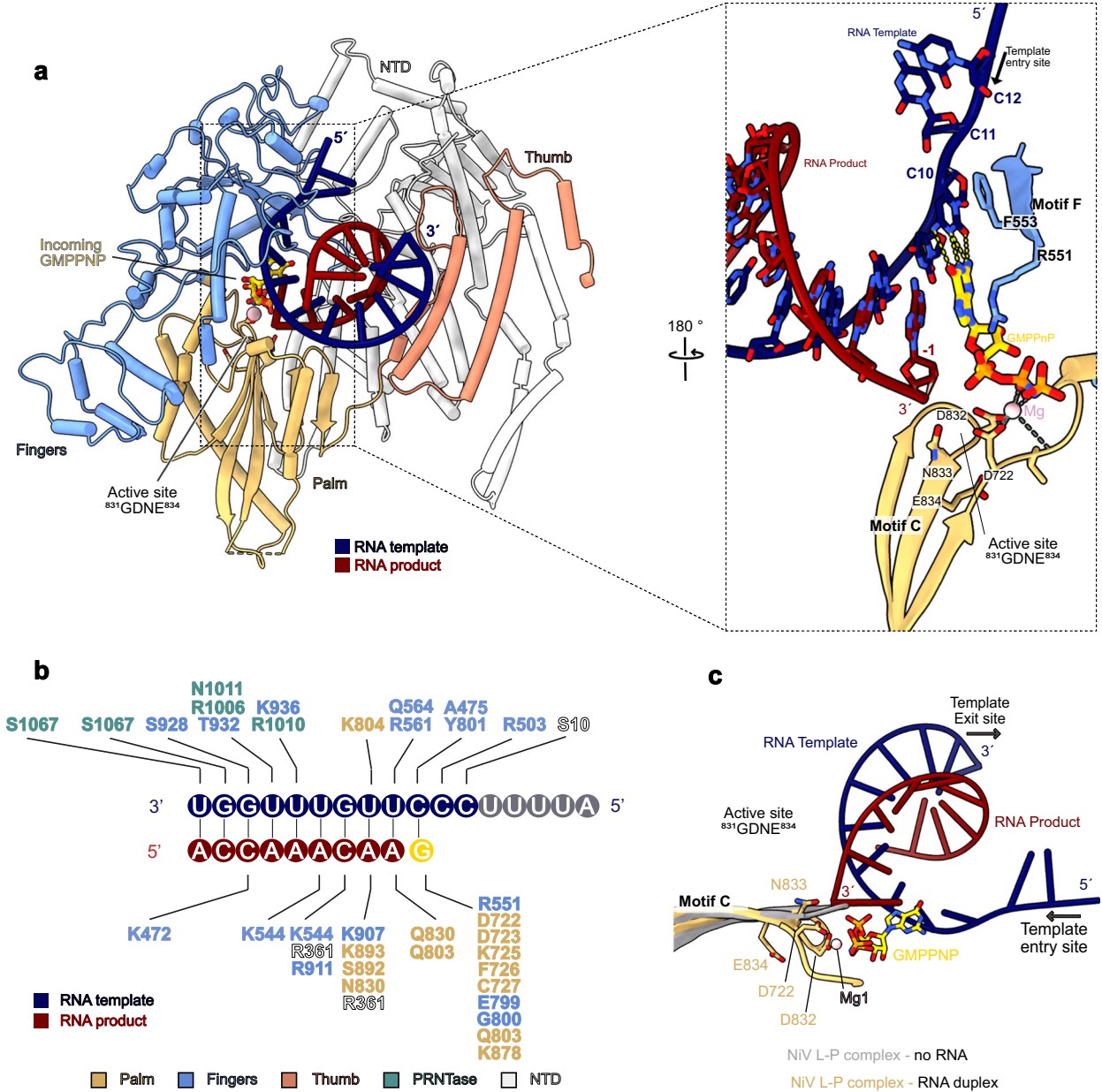

**Fig. 3 | Details of interactions between the NiV L-P complex and RNA. a** Overall fold of the RdRp domain with the fingers, palm, and thumb subdomains colored and labeled as indicated. A close-up view of the active site shows the interactions between the RNA duplex and incoming NTP with motifs C and F. **b** Schematic of protein-RNA interactions. **c** Superposition of motif C in the apo- and elongating NiV L-P structures reveals a conformational change in the active site to accommodate the duplex RNA.

capping. In the elongating NiV L-P complex, this potential RNA path appears obscured, as also observed in apo hPIV3 and hPIV5 L-P structures (Supplementary Fig. 8b). Thus, the structure of the elongating NiV L-P complex reveals the orientation of the flexible CD-MTase-CTD module during early elongation, but additional rearrangements may occur upon further elongation of the product RNA and transitioning of the polymerase to either transcription or replication.

## Conformational rearrangements upon RNA binding and synthesis

Comparison between our two structures reveals conformational rearrangements that occur during the transition from the apo to the RNA-bound state of the NiV L-P complex. In the RNA-bound state, the priming loop (residues 1266–1290) and intrusion loop (1337–1362) become partially ordered (Fig. 5a). Both loops run along the interface between the RdRp and PRNTase domains and mediate interactions

between the two as well as the CD (Fig. 5b). In addition, both loops traverse the active site cavity of the RdRp domain and are located close to the upstream edge of the RNA duplex and the 5′ end of the nascent RNA. The intrusion loop remains extruded from the RdRp active site, thus leaving space for the RNA product and template to fit in. Intriguingly, in both loops, the parts in direct proximity to the putative RNA exit channel (res. 1276–1281 of the priming loop and 1342–1355 of the intrusion loop) show weak density, indicating that they remain flexible. This includes the HR motif in the intrusion loop, which interacts with the 5′ end of the RNA during capping[31]. While this results in a seemingly unobstructed tunnel for the RNA to exit towards the solvent, it is possible that the disordered regions occupy this path. Thus, further elongation of the RNA may trigger rearrangements in the enzyme. The comparison also reveals additional structural changes that occur in the RdRp domain of NiV L. In particular, the supporting loop becomes fully ordered and adopts a different conformation, and the supporting

helix, which is mobile in the apo structure, exhibits clear ordered density in the RNA-bound state (Fig. 5c and Supplementary Fig. 3b). These elements interact with the template RNA, the fingers domain and with the shifted motif C (Fig. 3c and Supplementary Fig. 5f). The supporting helix also interacts with and may stabilize the priming loop as well as the CD domain (Fig. 5c).

These rearrangements in NiV L may explain how RNA-binding leads to the ordering of the flexible C-terminal domains. Several structural elements that undergo conformational changes or become stabilized upon RNA binding interact with the CD domain, in particular the intrusion loop, priming loop, fingers domain, supporting loop and helix, and PRNTase domain (Fig. 5b–d). These interactions likely stabilize the CD, which in turn may lead to the ordering of the MTase and CTD in the observed orientation. Consistent with this, other nsNSV L-P structures with ordered C-terminal domains also show ordered priming and intrusion loops as well as supporting loop and helices and similar conformations of motif C (Supplementary Fig. 8c).

Taken together, our data reveal structural rearrangements in the RdRp core upon RNA binding and early elongation and suggest how these rearrangements may trigger large-scale changes in the architecture of the enzyme.

### Model for NiV RNA synthesis

The structures of the apo and elongating NiV L-P complex allow us to propose a structural model for the early steps of Nipah virus RNA synthesis. First, the NiV L-P is in its apo state, with disordered C-terminal domains as well as priming and intrusion loops and supporting loop and helix (Fig. 6a). Next, it binds to the leader RNA to form an initiation complex (Fig. 6b). Structural modeling based on the recently reported EBOV[16] and RSV[17] initiation complexes shows that no major structural rearrangements are necessary for this, as the template strand can be accommodated in the active site without clashes (Supplementary Fig. 6 f). Whether or not the priming loop transiently occupies the RdRp active site to facility initiation remains unclear. After initiation, the complex transitions into an early elongation state (Fig. 6c). In this state, the supporting loop and helix are fully ordered, and the priming loop and intrusion loop are partially ordered, which leads to the stable association of the previously mobile C-terminal domains. Whether these ordering events occur already during initiation or only after the synthesis of several nucleotides of product RNA is not clear. In this state, the complex can synthesize at least 9 nt of product RNA. Whether further elongation of the RNA then triggers additional conformational changes in the complex and how the ensuing events differ between replication and transcription remains to be determined.

### Discussion

Here, we present structures of the NiV L-P complex in its apo and actively elongating state. These structures not only provide molecular insights into the RNA synthesis machinery of Nipah virus but also represent the first structural snapshot of an actively elongating nsNSV L-P complex.

The apo structure reveals the architecture of the Nipah virus RNA polymerase. It shows that NiV L and four copies of NiV P associate to form a complex which adopts an overall structure that is highly conserved among nsNSV L-P complexes. Its core is formed by the L protein from which a rod-like stalk formed by the OD domains of four copies of P protrudes. The P-stalk of the NiV L-P complex exhibits a unique mushroom-like tip formed by helical regions that fold back onto the OD domains, which was also observed in a previous crystal structure of NiV P[32]. Our structure confirms that it also forms in the context of the intact L-P complex. While the RdRp and PRNTase domains of L are clearly resolved, the C-terminal CD, MTase, and CTD are invisible in the apo structure of the NiV L-P complex, suggesting conformational flexibility. This is consistent

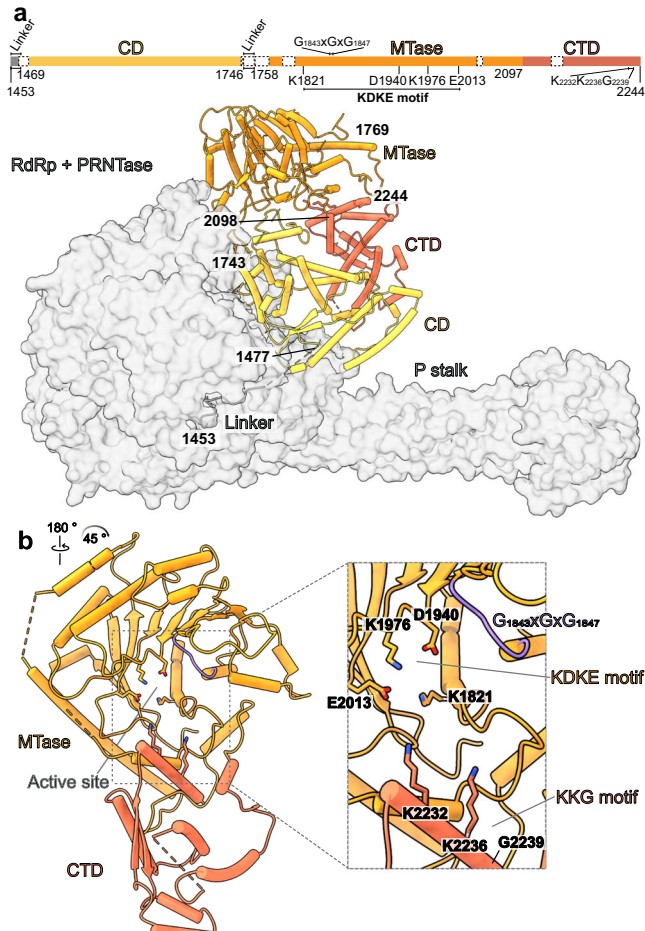

**Fig. 4 | Structure of the NiV L C-terminal domains. a** Schematic domain representation of the NiV L C-terminal domains with key catalytic residues indicated. Unmodelled regions are indicated as white dashed boxes. The CD, MTase, and CTD domain are shown as cartoon while the RdRp, PRNTase and P-stalk are shown as transparent gray surface. **b** MTase-CTD domain with a close-up view of the MTase active site. Arrows indicate rotation relative to the view in panel a. The KDKE and KKG motifs are shown as sticks, and the GxGxG motif is highlighted in purple.

with previously reported apo structures of RSV, HMPV, and EBOV L-P complexes, in which the C-terminal domains were also not visible[10–12]. By contrast, in the apo structures of VSV, NDV, hPIV3, and PIV5 L-P complexes, these domains were observed to be ordered, but adopt varying orientations with regards to the core[9,13–15,18]. Based on these structures, it has been suggested that the conformational state of the C-terminal domains correlates with distinct functional states of the L-P complex and is coupled to the conformation of the priming and intrusion loops[6,13]. In particular, in many structures with ordered C-terminal domains, either the priming or intrusion loops invade the active site cavity and would likely clash with an RNA duplex[8,9,13–15,19]. By contrast, in enzymes with disordered C-terminal domains, such as RSV, HMPV, or EBOV L-P complexes, the priming and intrusion loops are either retracted to the PRNTase domain or disordered, and the active site is unobscured. Consistent with this, in the apo NiV L-P complex structure, both the priming loop and the intrusion loop are disordered, and the C-terminal domains are flexible. During the preparation and revision of this manuscript, several other studies were published that also report structures of the apo NiV L-P complex, which are virtually identical to the one presented here[21–25]. In all cases, the CD, MTase, and CTD are also not visible, independently confirming that these domains are flexible in the apo state of the enzyme.

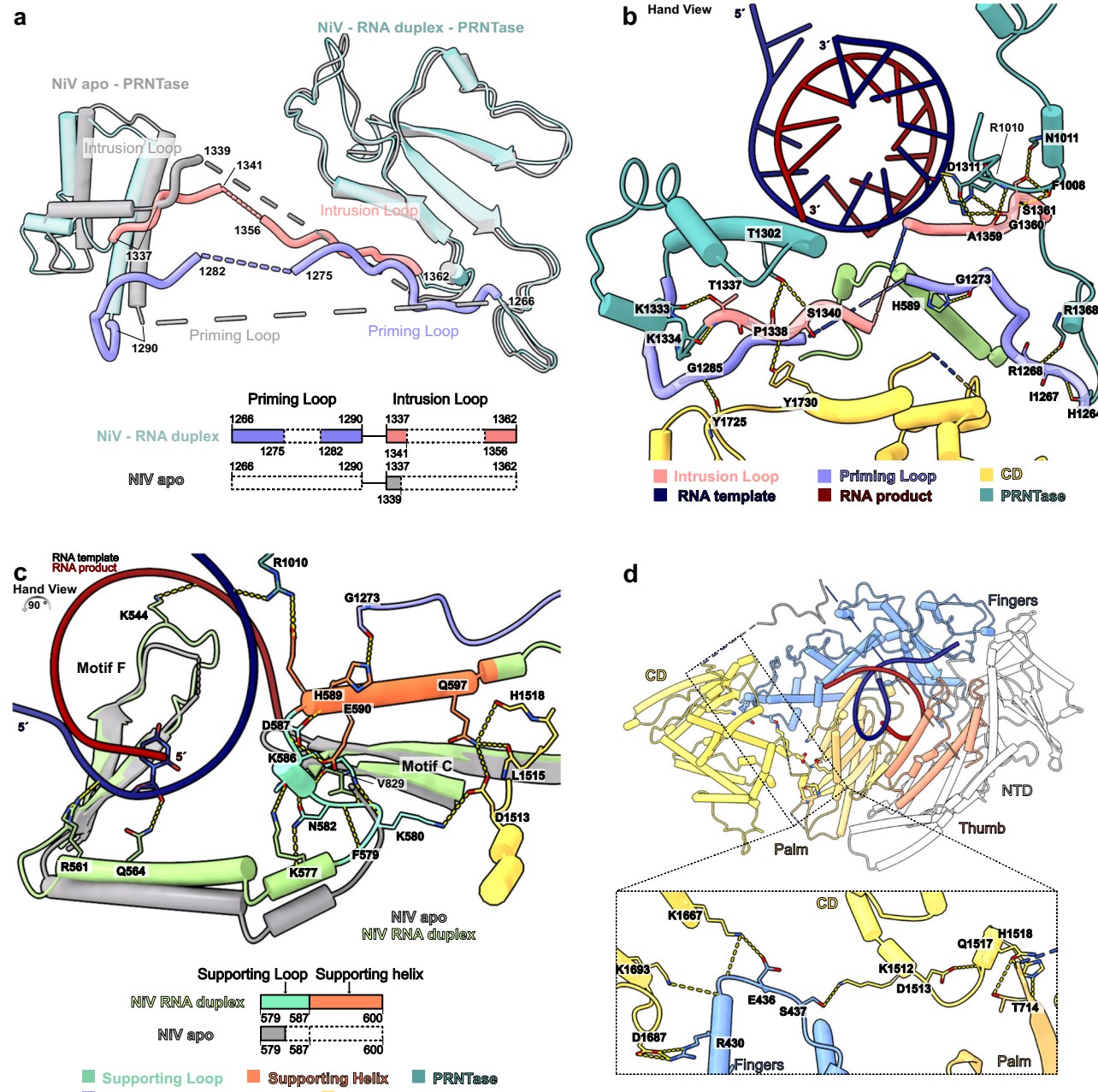

**Fig. 5 | Rearrangements in NiV L-P complex upon RNA binding. a** Structural and schematic comparison of the priming and intrusion loops in the apo- and elongating NiV L-P complex structures. Both loops become partially ordered in the elongating NiV L-P complex with an RNA duplex bound. **b** Detailed view of the interactions between the intrusion and priming loops and other elements in the elongating NiV L-P complex. Key interacting side and main chains are depicted as sticks, with hydrogen bonds highlighted by yellow dashed lines. **c** Structural overlay and schematic comparison of the supporting helix and loop in the apo (gray) and elongating NiV L-P complex (colored). The supporting loop and helix become ordered in the elongating RNA-bound NiV L-P complex. Interactions involving the supporting helix and loop are shown as yellow dashed lines. **d** Cartoon representation of the RdRp and CD domains in the elongating NiV L-P complex, with a close-up view of the interactions between them. Hydrogen bonds are depicted as yellow dashed lines and interacting residues are shown as sticks.

The structure of the RNA-bound NiV L-P complex represents the first structure of an nsNSV L-P complex in the actively elongating state. By using a GTP-analog with a non-hydrolyzable bond between the β and γ phosphate groups, we were able to trap an early elongation complex after in situ synthesis of 9 nt of product RNA and with the incoming substrate NTP bound in the + 1 site. The structure shows how the enzyme interacts with template and product RNA during early elongation and reveals key residues involved in substrate binding and catalysis. Structural comparisons show that the overall organization of the active site during RNA synthesis is highly conserved across RNA viruses. Our structure of the actively elongating NiV L-P complex thus provides an important puzzle piece for the mechanistic understanding of nsNSVs and a framework for studying the mechanism of action of nucleoside analog inhibitors.

Comparison between the two NiV L-P complex structures reported here reveals the structural rearrangements that accompany the transition from the apo to the RNA-bound state of the enzyme. Upon RNA binding and early elongation, the CD, MTase, and CTD become ordered. This is accompanied by rearrangements in the RdRp domain (supporting loop and helix) as well as the PRNTase domain (priming

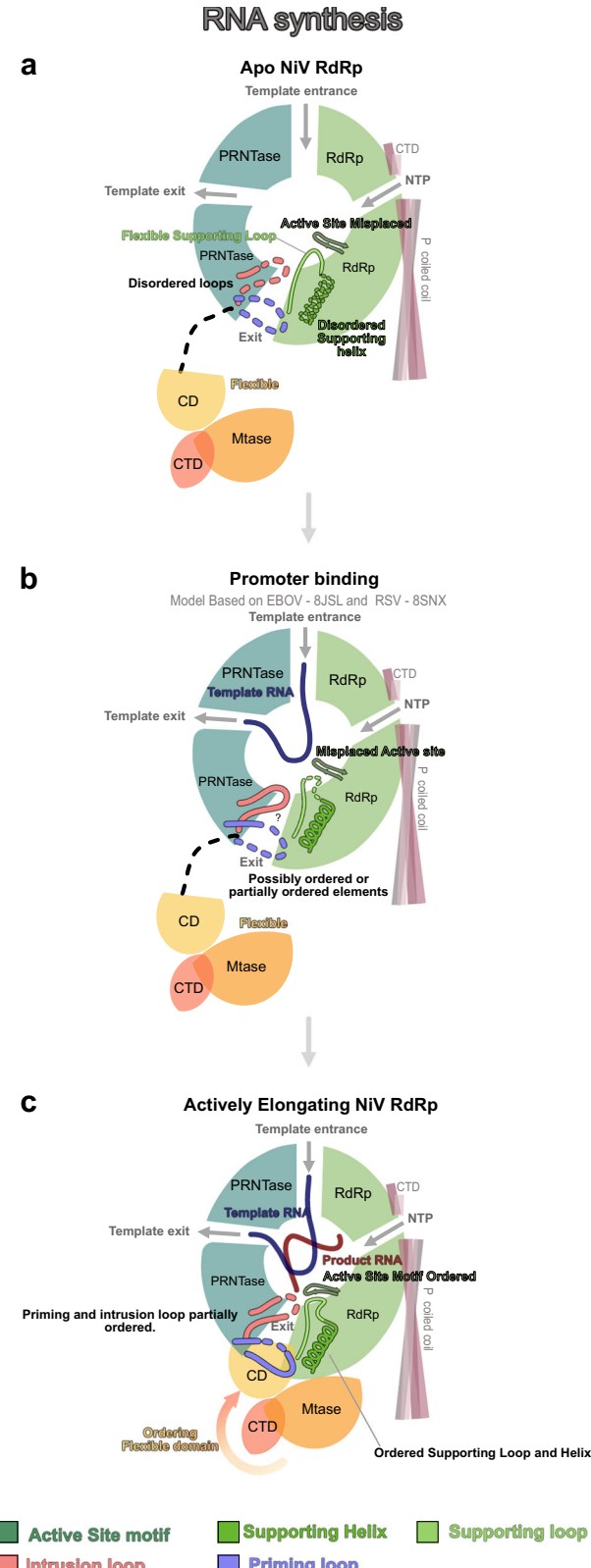

**RNA synthesis**

**a** Apo NiV RdRp

**b** Promoter binding
Model Based on EBOV - 8JSL and RSV - 8SNX

**c** Actively Elongating NiV RdRp

■ **Active Site motif**   ■ **Supporting Helix**   ■ **Supporting loop**
■ **Intrusion loop**      ■ **Priming loop**

**Fig. 6 | Structure-based model for early steps of NiV replication. a** In the **apo state**, the active site motif C is displaced, and the intrusion loop, priming loop, and supporting helix are disordered. The C-terminal domain is flexible. **b** Model of the **promoter bound state**. The model is based on structures of Ebola virus (EBOV)[16] and Respiratory Syncytial Virus (RSV)[17] L-P complexes in complex with promoter RNA (PDB IDs: 8JSL and 8SNX, respectively). No major rearrangements are required to accomodate the template RNA. The supporting loop, supporting helix, and intrusion loop may become partially or transiently stabilized. **c** In the **actively elongating state**, motif C shifts to accommodate the RNA product, and the supporting helix and loop become fully ordered. The priming and intrusion loops become partially ordered. The supporting loop and helix interact with motif C, the CD, the PRNTase domain, and the intrusion loop. The C-terminal domains become stabilized and ordered.

shown to be stabilized by interaction with motif C but no continuous density was observed for the supporting loop[10]. In the elongating NiV L-P complex, the priming and intrusion loops further become partially ordered and run along the interface between the RdRp and PRNTase domains, near the RNA exit tunnel. Both elements interact with the CD domain and, in conjunction with the supporting loop and helix, likely support its ordering, suggesting a mechanism how RNA-binding in the RdRp core could lead to the ordering of the flexible C-terminal domains. Thus, the structure of the elongating NiV L-P complex reveals the architecture of its previously mobile C-terminal domains and suggests how structural rearrangements during early elongation led to their ordering.

The L-P complex is responsible for the replication and transcription of the viral genome. In both cases, RNA synthesis is initiated de novo at the 3' end of the (-) sense genome, but the ensuing steps differ. In the case of replication, the L-P complex must transition into a productive elongation state to produce a full-length, 5'-triphosphorylated copy of the (−) sense genome. By contrast, in the case of transcription, it must re-initiate at one of several downstream gene start sites and subsequently carry out 5' capping of the nascent transcript. The structure of the RNA-bound NiV L-P complex presented here represents an early elongation state, in which only 9 nt of RNA have been synthesized and in which the polymerase has likely not yet committed to one of the two processes. As the product elongates, further conformational changes may occur, which may ultimately determine whether the complex transitions into a transcriptional or replicative state. How the different orientations of the CD-MTase-CTD module observed in different L-P complex structures correlate to functional states thus needs to be further investigated.

In summary, our data reveal the structural basis of Nipah virus RNA synthesis and provide mechanistic insights into nsNSV replication and transcription in general. This provides a framework for understanding the mechanisms of replication and transcription of a broad range of human pathogens and may aid in the development of targeted anti-viral therapies.

## Methods

No statistical methods were used to predetermine the sample size. The experiments were not randomized, and the investigators were not blinded to allocation during experiments and outcome assessment.

### Cloning and protein expression

Sequences encoding NiV L and NiV P (NCBI Reference Sequence NC_002728.1) were obtained as codon-optimized variants for expression in insect cells from GeneArt. The sequence encoding NiV L was cloned into the 438-C vector (kind gift from Scott Gradia; Addgene plasmid # 55220; http://n2t.net/addgene:55220; RRID:Addgene_55220[43]) using ligation-independent cloning (LIC). The construct was designed in frame with an N-terminal 6xHis tag followed by a maltose binding protein and a TEV cleavage site (forward primer 5'-TACTTCCAAT CCAATGCAGCCGACGAACTGTCTATCTCCG-3' (IDT) and reverse primer

and intrusion loops). In the apo state of the NiV L-P complex, the supporting loop and helix, as well as the priming and intrusion loops are partially ordered or disordered. In the elongating NiV L-P complex, the supporting loop and helix become fully ordered and are stabilized by interactions with the RNA and the RdRp domain as well as with the intrusion loop and CD. This is reminiscent of a previous structure of EBOV RdRp incubated with RNA, in which the supporting helix was also

5′-TTATCCACTTCCAATGTTATTAGATGATGGAGATGTAGCCGATGATC
TTCC-3′ (IDT)). Similarly, the sequence encoding NiV P was cloned into
the 438-A vector (kind gift from Scott Gradia; Addgene plasmid # 55218;
http://n2t.net/addgene:55218; RRID: Addgene_55218[43]) encoding a
tagless variant (forward primer 5′-TACTTCCAATCCAATCGATGG
ACAAGCTGGAACTGGTCAAC-3′ (IDT) and reverse primer 5′-TTATCC
ACTTCCAATGTTATTAGATGTTGCCGTCAATGATGTCGTTG-3′ (IDT)).
Both vectors were used for subcloning a co-expression vector encoding
the 6xHis-MBP-NiV-L and NiV-P proteins. The sequences were confirmed
by Oxford Nanopore Technology (ONT) sequencing (Microsynth Seq-
lab). Proteins were expressed in an insect cell expression system as
previously described[44,45]. Briefly, V0 bacmids were extracted from
positive clones using isopropanol precipitation and transfected into Sf9
cells cultured in ESF921 medium (Expression Technologies) with the
X-tremeGENE9 transfection reagent (Sigma). The V0 virus was harvested
72–120 hours post-transfection and used to generate V1 by infecting Sf9/
Sf21 cells. After 72 h of incubation at 27 °C with shaking, the V1 virus was
collected and stored at 4 °C. For large-scale protein expression, Hi5 cells
were cultured in an ESF921 medium and infected with the V1 virus. The
culture was incubated at 27 °C for 72 h before harvesting the cells by
centrifugation.

### Protein purification

After growth, cells were harvested by centrifugation and resus-
pended in Buffer A (50 mM HEPES pH 8.0, 400 mM NaCl, 6 mM
MgCl$_2$, 10% glycerol, 5 mM DTT, and 0.01% Tween 20), supple-
mented with cOmplete™, EDTA-free Protease Inhibitor Cocktail
(Sigma-Aldrich). Cell lysis was performed by sonication, followed by
centrifugation at 25,000 rcf for 1 h at 4 °C. The soluble fraction was
collected and subjected to ultracentrifugation at 235,000 rcf for
60 min at 4 °C. The supernatant was filtered through 0.8 μm mem-
branes and applied to a HisTrap HP 5 mL column (Cytiva) pre-
equilibrated with Buffer A. The column was washed with 10 column
volumes (CV) of Buffer A, followed by 5 CV of high-salt buffer (Buffer
A with additional 600 mM NaCl). The protein was eluted using
Buffer A containing 500 mM Imidazole. For further purification, the
eluate was incubated with amylose beads for 2 h at 4 °C. The resin
was loaded onto an empty column and washed with 10 CV of Buffer A
to remove unbound proteins. The protein of interest was eluted with
Buffer A containing 200 mM maltose. The eluted protein was
digested with 1 mg of His-tagged TEV protease (12 h at 4 °C). The
digested protein was reapplied to a HisTrap column equilibrated with
Buffer A. The flow-through was diluted to a final NaCl con-
centration of 150 mM (Buffer B: 50 mM HEPES pH 8.0, 150 mM NaCl,
6 mM MgCl$_2$, 10% glycerol, 5 mM DTT, and 0.01% Tween 20) and
applied to a HiTrap Heparin 5 mL column (Cytiva) equilibrated with
Buffer B. The protein was eluted with 300 mM NaCl using a gradient
from 150 to 2000 mM. Peak fractions, as identified by SDS-PAGE,
were pooled and concentrated using a MWCO 100,000 Amicon
Ultra Centrifugal Filter (Merck). The concentrated sample was
applied to a Superose 6 10/300 column (Cytiva) equilibrated with
Buffer A. Fractions containing the Nipah L-P complex were con-
centrated again using a MWCO 100,000 Amicon Ultra Centrifugal
Filter (Merck). The complex with a final concentration of 1.5 μM
was freshly used for grid preparation and/or aliquoted and flash-
frozen and stored at 80 °C until use. Protein identity was assessed by
SDS-PAGE and confirmed by mass spectrometry (Supplemen-
tary Fig. 1b).

### Mass spectrometry

Protein bands were excised from the SDS-PAGE gel, washed, reduced
with dithiothreitol (DTT), alkylated with iodoacetamide and digested
with trypsin (sequencing grade, Promega) overnight. The resulting
peptides were extracted, dried in a SpeedVac vacuum concentrator

(Thermo Scientific) and dissolved in 2% acetonitrile/0.05% tri-
fluoroacetic acid (v:v).

Peptides were analyzed by electrospray ionization mass spectro-
metry in a Thermo Orbitrap Exploris 480 mass spectrometer coupled
to an UltiMate3000 ultrahigh performance liquid chromatography
system (Thermo Scientific) with an in-house packed C18 reverse-phase
column (75 μm ID × 300 mm, Reprosil-Pur 120 C18-AQ, 3 μm, Dr.
Maisch). The mass spectrometer was equipped with a Nanospray Flex
Ion source and controlled by Thermo Scientific Xcalibur 4.4 software.
Data were acquired using a Top30 data-dependent acquisition
method. One full MS scan across the 350–1400 $m/z$ range was acquired
at a resolution of 60,000, with an AGC target of 300% and a maximum
fill time of 25 ms. Precursors with charge states 2–6 above a 1e4
intensity threshold were then sequentially selected using an isolation
window of 1.6 m/z, fragmented with nitrogen at a normalized collision
energy setting of 28%, and the resulting MS2 spectra recorded at a
resolution of 15000, AGC targets of 50% and a maximum fill time of
50 ms. Dynamic exclusion of precursors was set to 22 s.

Proteins were identified with MaxQuant 2.6.1.0[46] by searching
Thermo raw files against a database that included sequences of over-
expressed Nipah virus proteins, *Trichoplusia ni* UniProt reference pro-
teome (release 29-05-2024) and common contaminants observed in MS
experiments. Two missed cleavages were allowed. C-carbamidomethyl
as well as protein N-terminal acetylation and M-oxidation were set as a
fixed and variable modifications, respectively.

### RNA synthesis assay

The RNA template derived from the NiV leader promoter
(3′-UGGUUUGUUCCCUUUUA-5′) was purchased from Integrated
DNA Technologies. The reaction mixture consisted of 20 μM NiV lea-
der, 1.2 μM recombinant L-P complex, and a buffer containing 50 mM
HEPES (pH 8.0), 5 mM DTT, 10% glycerol, and 6 mM MgCl$_2$. Reactions
were incubated at 30 °C for 10 min and then initiated by the addition of
nucleotides (1 mM ATP (Thermo Scientific, R0441), 150 μM fluorescein-
labeled CTP (Jena Bioscience, NU-831-FAMX), and 150 μM GMPPNP
(Sigma, G0635) in a final volume of 10 μL, followed by incubation at
30 °C for 3 h. Control reactions were performed in the absence
of NTPs.

The reactions were stopped by adding a buffer containing 7 M
urea, 50 mM EDTA (pH 8.0), and 1x TBE buffer. The samples were then
treated with 6 U of proteinase K (New England Biolabs) at 37 °C for
30 min, denatured at 95 °C for 10 min, and analyzed on a 20% poly-
acrylamide denaturing urea gel. The migration products were visua-
lized using a Typhoon phosphorimager (GE Healthcare). The lengths of
the RNA products were determined by comparing them to a ladder of
synthesized FAM-labeled RNA molecules of known lengths.

### Cryo-EM sample preparation and data collection

For the apo NiV L-P complex, 0.8 μM of freshly prepared Nipah L-P
protein were used for cryo-EM grid preparation. For the elongating NiV
L-P complex, 0.6 μM Nipah L-P protein in Buffer B were mixed with a 15-
fold molar excess of RNA scaffold and 1 mM ATP (Thermo Scientific,
R0441), 1 mM CTP (ThermoScientific, R0451) and 150 μM GMPPNP,
followed by incubation on ice for 2 h prior to cryo-EM grid preparation.
For both samples, 3.5 μL were applied to freshly glow-discharged R 2/1
holey carbon grids (Quantifoil) at 4 °C and 95% humidity. The grids
were blotted for 5 s with a blot force of 5 using a Vitrobot Mark IV
(Thermo Fisher Scientific) before vitrification in liquid ethane.

Cryo-EM data collection was performed with SerialEM[47] using a
Titan Krios transmission electron microscope (Thermo Fisher Scien-
tific) operated at 300 keV. Images were acquired in EFTEM mode with a
slit width of 20 eV using a GIF quantum energy filter and a K3 direct
electron detector (Gatan) at a nominal magnification of 105,000x
corresponding to a calibrated pixel size of 0.834 Å/pixel. For the apo

NiV L-P complex data set, exposures were recorded in counting mode for 1.48 s with a dose rate of 23 e⁻/px/s resulting in a total dose of 48.94 e⁻/Å² that was fractionated into 50 movie frames. For the elongating NiV L-P complex data set, exposures were recorded in counting mode for 2.14 s with a dose rate of 16.9 e⁻/px/s resulting in a total dose of 52.00 e⁻/Å² that was fractionated into 50 movie frames.

### Cryo-EM data processing and analysis

Motion correction, CTF estimation, particle picking, and extraction were performed on the fly using Warp[48].

An overview of the cryo-EM processing workflow for the apo NiV L-P complex dataset is depicted in Supplementary Fig. 2. Raw micrographs were split into four batches for initial processing. Particles were extracted with a box size of 480 px and binned threefold. All processing steps were carried out in Cryosparc v.4.5.3[49] unless stated otherwise. Particles from the first two batches were subjected to initial 2D classification, and 'good' and 'junk' classes were selected manually, and each used for ab initio model generation. The resulting best model from the 'good' particles and four 'junk' models were then used as input references for a supervised heterogeneous refinement job with all particles. To obtain an improved reference model, particles belonging to the 'good' class (~36%; 3.6 million particles) were then again subjected to 2D classification with subsequent ab initio model generation from the 'good' and 'junk' classes. The resulting best model from the 'good' particles and the 'junk' models were then used as input for a supervised heterogeneous refinement job with the unbinned particles from the good class of the first heterogeneous refinement. Particles belonging to the 'good' class (~71%, 2.5 million particles) were then subjected to unsupervised heterogeneous refinement with 5 classes. Particles belonging to the best class (~22%, 575 thousand particles) were subjected to non-uniform refinement, which resulted in a reconstruction at a nominal resolution of 2.5 Å that showed substantial directional anisotropy most likely due to limited view distribution. This map was used as a reference together with the models from the first 'junk' ab initio job for a supervised heterogeneous refinement using the unbinned particles from the good class of the first supervised heterogeneous refinement as input. Particles belonging to the 'good' class (~70%; 2.5 million particles) were subjected to another round of supervised heterogeneous refinement (4 classes). Particles belonging to the best class (~24%; 590 thousand particles) were exported to Relion 5.0[50] and subjected to 3D refinement with blush regularization enabled[51]. This resulted in an isotropic reconstruction at an overall resolution of 2.6 Å after post-processing.

An overview of the cryo-EM processing workflow for the RNA-bound NiV L-P complex dataset is depicted in Supplementary Fig. 5. Raw micrographs were split into four batches for initial processing. Particles were extracted with a box size of 480 px. All processing steps were carried out in Cryosparc v.4.5.3[49] unless stated otherwise. Particles from the first batch were subjected to initial 2D classification and 'good' and 'junk' classes were selected manually and each used for ab initio model generation. The resulting best model from the 'good' particles and five 'junk' models were then used as input references for supervised heterogeneous refinements carried out for each batch separately (except for batch 1 + 2, which were processed together). Particles belonging to the 'good' class (~34%) were subjected to another round of supervised heterogeneous refinement with one 'good' and five 'junk' models, and the particles belonging to the 'good' class from each batch (~71%) were pooled and subjected to unsupervised heterogeneous refinement with six classes. This led to the clear separation of particles with and without density for the C-terminal flexible domains of NiV L. The particles belonging to the best class with the C-terminal domains ordered (~35%, 1 million particles) were exported to Relion 5.0[50] and subjected to global 3D refinement with blush regularization followed by 3D classification without image alignment, T value of 100, $K = 6$ and a mask around the NiV L core. Particles belonging to the class with the best density for the RNA duplex in the active site were selected (~33%; 331 thousand particles) and used for 3D refinement with blush regularization[51]. This led to a consensus reconstruction of the RNA-bound NiV L-P complex at an overall resolution of 2.8 Å. To further improve the resolution in flexible parts of the NiV L-P complex, MultiBody refinement using three masks (RdRp core, P-stalk, and CTD) was carried out, which led to focused maps of these regions with 2.8, 3.6, and 3.1 Å resolution, respectively, and improved the density for the distal parts of the P-stalk substantially. The focused maps were combined into a composite map using the vop maximum command in Chimera[52].

### Model building and refinement

To generate initial models, the structures of NiV L and P proteins were predicted using Alphafold 3[53]. The predicted P protein structure was then partially replaced in the model with the crystal structure of NiV P (PDB ID: 4N5B)[32] while retaining the XD domain from the Alphafold model. The overall model was subsequently fitted into the maps as a rigid body using UCSF ChimeraX[54] and manually rebuilt using Coot[55]. Refinement was carried out by real-space refinement in Phenix[56,57] as well as by molecular dynamics assisted refinement using ISOLDE[58]. For the Apo NiV L-P complex, final refinement was carried out using the local resolution filtered map. For the elongating NiV L-P complex, final refinement was carried out using the composite map followed by a single round of ADP refinement against the consensus map. The resulting models were validated using the MolProbity package within the Phenix suite[59]. Refinement statistics are shown in Supplementary Table 1. Figures were prepared with ChimeraX[54].Cavities in the complex were calculated using CAVERweb[60]. Multiple sequence alignments were performed with MultAlin[61] and results visualized with ESPript3[62].

### Reporting summary

Further information on research design is available in the Nature Portfolio Reporting Summary linked to this article.

## Data availability

The electron potential reconstructions were deposited with the Electron Microscopy Database (EMDB) under accession codes EMD-51402 (apo NiV L-P complex) and for the elongating NiV L-P complex: EMD-51403 (composite map), EMD-51723 (Map 1 - RdRp core), EMD-51724 (Map-2 – P protein), EMD-51725 (Map 3 – L protein C-terminal domains) and EMD-51722 (consensus refinement map). The structure coordinates were deposited to the Protein Data Bank (PDB) under accession codes 9GJT (apo NiV L-P complex) and 9GJU (elongating NiV L-P complex). Source data are provided in this paper.

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

## Acknowledgements

We thank all members of the Hillen Lab for discussions and Christian Dienemann and Ulrich Steuerwald (MPI-NAT cryo-EM facility) for assistance with cryo-EM data acquisition. H.S.H. was supported by the Deutsche Forschungsgemeinschaft under Germany's Excellence Strategy EXC 2067/1-390729940, FOR2848 (P10), SFB1190 and SFB1565 (Project number 469281184, P13), by the European Union (ERC Starting Grant MitoRNA, grant agreement no. 101116869) and by the EMBO Young Investigator Program. Views and opinions expressed are, however, those of the author(s) only and do not necessarily reflect those of the European Union or the European Research Council Executive Agency. Neither the European Union nor the granting authority can be held responsible for them. We acknowledge support by the Open Access Publication Funds/transformative agreements of the Göttingen University.

## Author contributions

F.A.S. designed and carried out all experiments and data analysis unless stated otherwise. K.D. cloned NiV L and P and assisted with insect cell culture. O.D. and H.U. carried out mass-spectrometric analysis. H.S.H. designed and supervised research. F.A.S. and H.S.H. interpreted data and wrote the manuscript.

## Funding

## Competing interests

The authors declare no competing interests.
