## [Transparent Peer Review file · Nature Communications]

Structural basis of Nipah virus RNA synthesis

Corresponding Author: Professor Hauke Hillen

Version 0:

Reviewer comments:

Reviewer #1

(Remarks to the Author)

In this study, the authors resolved two high-resolution structures of the NiV L-P complex in its apo and early-elongation states. The latter one represents a structural snapshot of an actively elongating L-P complex, with template RNA, product RNA, and incoming NTP bound, which is the highlight of the study. This finding helps enrich the understanding of nsNSV L-P complex during the RNA synthesis.

Major points:

1. Replication or transcription. The key finding of this work is to resolve the structure of NiV L-P complex with template RNA, product RNA, and incoming NTP bound. The authors claimed this state as the "replication" state throughout the manuscript including the title. Is there any evidence to strongly support this claim? The biochemical assay provided in the manuscript can only be claimed as "RNA synthesis". The difference between genome replication and transcription mainly lies in the channel connectivity to the MTase and the RNA coated by nucleoprotein during replication. It will be great if authors compare the resolved structures with other nsNSV L-P complexes following the reference (Li, T. et al. Structures of the mumps virus polymerase complex via cryo-electron microscopy. *Nat. Commun.* 15, 4189 (2024)).
2. RNA in the apo-state NiV L-P complex. In the extended data Figure 5, authors also found the apo-state NiV L-P complex (the ratio at 32%) in the whole dataset. It is not clear whether there is RNA, no matter template RNA or product RNA, resolved in this structure. If not, RNA may play an important role in determining more density. Can authors increase the ratio of NiV L-P with the appendage (CTD map) via elevating RNA ratio?
3. Nomenclature. It is confused to use NiV L-P complex as NiV RdRp complex, since L has a RdRp domain. The same nomenclature issue occurs on P. Four P molecules in P tetramer are usually depicted as P1, P2, P3, and P4, respectively. It is better to keep consistent with other papers in the field.
4. Line 22/350: The article of "Yang, G., Wang, D. & Liu, B. Structure of the Nipah virus polymerase phosphoprotein complex. *Nat Commun* 15, 8673 (2024)" has been published, please re-evaluate the contribution of this manuscript.

Minor points:

1. The four copies of the P protein are difficult to distinguish due to their similar colors. To facilitate comparison with other nsNSVs, it is recommended to label them using other color strategy.
2. In Extended Data Figure 3a and Figure 2e, the yellow label "motif C" should be corrected to "motif D".
3. In Figure 3B and Extended Data Figure 6a, the term "RNA template" should be in blue.
4. Line 603: It should specify "20% polyacrylamide."

Other writing formats:

1. "Ptcls" should be changed to "particles" (Extended Data Figure 2b).
2. "GMPPnP" should be corrected to "GMPPNP" (Figure 3a and manuscript).
3. "MUMPS" should be revised to "MUV" (Extended Data Figure 7).
4. "Nipah" should be revised to "Nipah virus" (Line 633).
5. "TEV" should be clarified as "TEV protease" (Line 720).
6. "Mumps" should be corrected to "Mumps virus".
7. "uM" should be corrected to "μM" (Line 730).
8. Units (min/minutes, h/hours) and spacing should be corrected (e.g., Lines 228/720/762/767).

Reviewer #2

(Remarks to the Author)

The work by Sala et al. presents cryoEM structures of the Nipah virus polymerase-phosphoprotein complex in both apo and early replicating states. This work is largely a structural description with excellent resolution of the RNA and bound nucleotide analog in the replicating polymerase structure. This is an important work that strongly contributes to our understanding of nsNS RNA virus replication.

The enthusiasm for the structure determination of Nipah polymerase-P is somewhat lessened by a competing paper that published a nearly identical apo structure earlier this fall. However, the inclusion of RNA and an incoming nucleotide and, in particular, the clarity of these substrates in the density will make this an impactful work.

Major comments:

Mass spec methods are included, but mass spec results are only briefly mentioned and actual data is missing.

Minor comments:

ED Fig 1c, please increase the font size the +/- symbols to a legible size.

Fig 2b. As I understand it, this is the composite map of several reconstructions and this should be made clear in the figure legend. The consensus map was not included among the requested maps.

ED Fig 3b, labels for colored domains would help orient readers

Fig 5a, the color choices for the protein are poor here and it is hard to distinguish the two structures

Please give suppliers for the nucleotide analogs used in this study

Reviewer #3

(Remarks to the Author)

Nipah virus is a non-segmented negative-strand RNA virus (nsNSV) with a high pandemic potential in humans. Its RNA-dependent RNA polymerase (RdRp) complex is made by the L protein, a large multi-functional polypeptide that contains all required enzymatic activities, and its cofactor the phosphoprotein (P). The complex replicates and transcribes the viral genome and is so the ideal target to design new specific antivirals. Recent years have seen a profusion of nsNSV RdRp structures, some incomplete, some complete, with or without RNA mimicking the genomic RNA, most of them looking inactive. With this manuscript, Sala and co-workers present a structural analysis of Nipah RdRp, in apo and replicative conformations that constitutes a major achievement in the field. In particular, they can observe the position of an incoming nucleotides for the RNA synthesis in the catalytic core of the enzyme that has never been observed in any previous work. The manuscript is well written, describes perfectly the observations and therefore deserves to be published in Nature Communications.

Different aspects should also considered before acceptance.

Major

Regarding the manuscript, my major criticism concerns the way the manuscript is written, in particular how the authors consider the works of Grimes et al. (DOI: 10.21203/rs.3.rs-4663080/v1) and Yang et al. (Nat Commun; PMID: 39375338), published on the same subject 2024 July 12th and October 7th respectively. It means that the summary of the paper is not true, i.e. "However, to date no structural data is available on the NiV RdRp complex" as well as different sentences in the main text. If the three manuscripts (this one and the two already published) had a similar timeframe, this could have been perfectly acceptable, but that is not the case. The paper proposed by the authors bringing much more details than the two others do (I fully appreciate the quality of the work done), so the authors should take full advantage of this situation. As an example, Yang et al. with their incomplete structure speculate vaguely on the role of D832 for the transcription, whereas in the present active structure, this residue appears central for the RNA synthesis process and so crucial for the whole activity of the complex. Several aspects of the manuscript should therefore be reviewed to take these facts into account (with proper references).

The authors use the abbreviation "CAP" to refer solely to the GDP polyribonucleotidyltransferase domain (please add GDP line 59) that initiates the capping process at the 5' of the viral mRNA. Considering the biochemical aspect, especially the cap being the guanosyl moiety (added by the PRNTase activity) as well as the methyl in position 7 of the ring (added independently from the first activity by the methylase domain), I think the use of this abbreviation can be confusing. Furthermore, considering also that the RdRp of segmented NSV such as Orthomyxoviridae and/or Bunyaviridae having a Cap-binding domain, this could so be a second source of discrepancy with the use of such trivial/easy-to-use abbreviation.

Minor

- Please, check the names/abbreviations of the journals in the references: Example for ref 9 & 15 with PNAS with or without ".". Idem ref 5 & 30 with J Virol with or without ".". Please homogenize all the references.
- all Latin-sounding virus family names (Paramyxoviridae, Filoviridae, ...) might be in italics
- homogenize "%" and "°C". X% and X °C (with space between numbers and %) as well as X°C and X °C can be found!
- The Greek letter "gamma" used looks more a "y" than a real "γ". Also doubts for the "β"

Version 1:

Reviewer comments:

Reviewer #1

(Remarks to the Author)

The manuscript has been significantly improved after revision. However, I do have two extra minors before the manuscript is appropriate for publication.

1. Line 480-481, the authors provided the supplier for GMPPNP, but the suppliers of other nucleotides, especially fluorescein-labelled CTP, are still not listed. In addition, the concentration of fluorescently-labeled CTP was reported as 150 mM, while the concentrations of other nucleotides were stated to be 1 mM. Authors need to double check the concentrations of all these nucleotides utilized in the manuscript.

2. Line 443-444, the authors state the peak fractions that eluted from the heparin column were identified by SDS/PAGE (Extended Data Figure 1b). The coming question is whether the sample used for mass spectrometry was also from the heparin column elution rather than from the size exclusion chromatography (line 91). The SDS-PAGE shows two Phosphoprotein bands (band2 band3 in Extended Data Figure 1b) were present in heparin column elution. Does the elution fraction of the SEC still have these two bands?

Reviewer #2

(Remarks to the Author)

My concerns have been addressed.

Reviewer #3

(Remarks to the Author)

Response to reviewers

Responses are in italique.

Changes are marked in red in the manuscript file.

Reviewer #1 (Remarks to the Author):

In this study, the authors resolved two high-resolution structures of the NiV L-P complex in its apo and early-elongation states. The latter one represents a structural snapshot of an actively elongating L-P complex, with template RNA, product RNA, and incoming NTP bound, which is the highlight of the study. This finding helps enrich the understanding of nsNSV L-P complex during the RNA synthesis.

We thank the reviewer for the positive evaluation and comments, which we addressed as outlined below.

Major points:

1. Replication or transcription. The key finding of this work is to resolve the structure of NiV L-P complex with template RNA, product RNA, and incoming NTP bound. The authors claimed this state as the “replication” state throughout the manuscript including the title. Is there any evidence to strongly support this claim? The biochemical assay provided in the manuscript can only be claimed as “RNA synthesis”. The difference between genome replication and transcription mainly lies in the channel connectivity to the MTase and the RNA coated by nucleoprotein during replication. It will be great if authors compare the resolved structures with other nsNSV L-P complexes following the reference (Li, T. et al. Structures of the mumps virus polymerase complex via cryo-electron microscopy. Nat. Commun. 15, 4189 (2024)).

We agree that our current biochemical data only demonstrate RNA synthesis and do not differentiate between replication or transcription. Since in both cases the L-P complex initiates RNA synthesis at the 3' leader sequence, the state we have resolved likely occurs during both processes before the polymerase has committed to one or the other. To reflect this, we now refer to this state as "actively elongating" throughout the manuscript and have changed the title of the paper accordingly.

As suggested by the reviewer, we have also included a comparison of the connectivity between the RdRp and MTase active sites in different nsNSV L-P complexes (new Extended Data Figure

8) and have extended the corresponding parts of the results section. In addition, we have also added a paragraph to the discussion in which we emphasize the early nature of the elongation complex resolved in this study.

2. RNA in the apo-state NiV L-P complex. In the extended data Figure 5, authors also found the apo-state NiV L-P complex (the ratio at 32%) in the whole dataset. It is not clear whether there is RNA, no matter template RNA or product RNA, resolved in this structure. If not, RNA may play an important role in determining more density. Can authors increase the ratio of NiV L-P with the appendage (CTD map) via elevating RNA ratio?

Indeed, we found that a substantial proportion of the particles in this dataset represent the apo state. The reconstruction from these is identical to the apo reconstruction from the first dataset, with no indications of either RNA template or product. To emphasize this, we now clearly indicated this in bold in Extended Figure 5.

Regarding the question of the RNA and extra density, we indeed observe ordered density for the C-terminal domain in all particles with RNA present. We used a 15-fold molar excess of RNA relative to the L-P complex during complex preparation for cryo-EM, but it is very well possible that the proportion of NiV L-P particles with ordered CTDs could be increased by incubating with an even larger excess of RNA. However, we did not explore this possibility experimentally as the number of particles we obtained with RNA bound was sufficient to generate a high-resolution reconstruction.

3. Nomenclature. It is confused to use NiV L-P complex as NiV RdRp complex, since L has a RdRp domain. The same nomenclature issue occurs on P. Four P molecules in P tetramer are usually depicted as P1, P2, P3, and P4, respectively. It is better to keep consistent with other papers in the field.

We agree with the reviewer that it is best to keep a consistent nomenclature across the literature. We have thus adopted the naming convention of labeling the four P molecules as P1, P2, P3, and P4 and refer to the complex as NiV L-P complex, in line with other publications.

4. Line 22/350: The article of “Yang, G., Wang, D. & Liu, B. Structure of the Nipah virus polymerase phosphoprotein complex. Nat Commun 15, 8673 (2024)” has been published, please re-evaluate the contribution of this manuscript.

We agree with the reviewer that this work needs to be accounted for. We would like to clarify that this paper was published after we wrote and first submitted our manuscript, and it was

thus not cited and discussed. We have revised our manuscript to reflect the current state of the literature, also including two further recently published papers on the apo NiV L-P complex (PMID 39661676, 39627254).

Minor points:

1. The four copies of the P protein are difficult to distinguish due to their similar colors. To facilitate comparison with other nsNSVs, it is recommended to label them using other color strategy.

We thank the reviewer for pointing this out. We have changed the coloring such that there is more contrast between the colors to more clearly distinguish between the P monomers.

2. In Extended Data Figure 3a and Figure 2e, the yellow label "motif C" should be corrected to "motif D".

We thank the reviewer for pointing this out and have corrected it.

3. In Figure 3B and Extended Data Figure 6a, the term "RNA template " should be in blue.

We thank the reviewer for pointing this out. We have correct the figure.

4. Line 603: It should specify "20% polyacrylamide."

We thank the reviewer for pointing this out and have corrected it.

Other writing formats:

1. "Ptcls" should be changed to "particles" (Extended Data Figure 2b).

We have changed this as suggested.

2. "GMPPnP" should be corrected to "GMPPNP" (Figure 3a and manuscript).

We have changed this as suggested.

3. "MUMPS" should be revised to "MUV" (Extended Data Figure 7).

We have changed the Figure as suggested.

4. "Nipah" should be revised to "Nipah virus" (Line 633).

We have revised this as suggested.

5. "TEV" should be clarified as "TEV protease" (Line 720).

We have changed this as suggested.

6. "Mumps" should be corrected to "Mumps virus".

We have changed this as suggested.

7. "uM" should be corrected to "μM" (Line 730).

We have corrected this.

8. Units (min/minutes, h/hours) and spacing should be corrected (e.g., Lines 228/720/762/767).

We have corrected this.

Reviewer #2 (Remarks to the Author):

The work by Sala et al. presents cryoEM structures of the Nipah virus polymerase-phosphoprotein complex in both apo and early replicating states. This work is largely a structural description with excellent resolution of the RNA and bound nucleotide analog in the replicating polymerase structure. This is an important work that strongly contributes to our understanding of nsNS RNA virus replication.

The enthusiasm for the structure determination of Nipah polymerase-P is somewhat lessened by a competing paper that published a nearly identical apo structure earlier this fall. However, the inclusion of RNA and an incoming nucleotide and, in particular, the clarity of these substrates in the density will make this an impactful work.

We thank the reviewer for their positive evaluation!

Major comments:

Mass spec methods are included, but mass spec results are only briefly mentioned and actual data is missing.

In the revised manuscript, we now include mass spec results in Extended Data Figure 1.

Minor comments:

ED Fig 1c, please increase the font size the +/- symbols to a legible size.

We have increased the font size as suggested.

Fig 2b. As I understand it, this is the composite map of several reconstructions and this should be made clear in the figure legend. The consensus map was not included among the requested maps.

We have revised the figure legend to clearly state that a composite map is shown. We apologize that the consensus map was not among the files provided, and have now uploaded it together with the remaining files if the reviewer wishes to inspect it.

ED Fig 3b, labels for colored domains would help orient readers

We have now included labels in the figure.

Fig 5a, the color choices for the protein are poor here and it is hard to distinguish the two structures

We have revised the colors in this figure to higher contrast to improve the interpretability.

Please give suppliers for the nucleotide analogs used in this study

We now include the supplier for the GMPPNP used.

Reviewer #3 (Remarks to the Author):

Nipah virus is a non-segmented negative-strand RNA virus (nsNSV) with a high pandemic potential in humans. Its RNA-dependent RNA polymerase (RdRp) complex is made by the L protein, a large multi-functional polypeptide that contains all required enzymatic activities, and its cofactor the phosphoprotein (P). The complex replicates and transcribes the viral genome and is so the ideal target to design new specific antivirals. Recent years have seen a profusion of nsNSV RdRp structures, some incomplete, some complete, with or without RNA mimicking the genomic RNA, most of them looking inactive. With this manuscript, Sala and co-workers present a structural analysis of Nipah RdRp, in apo and replicative conformations that constitutes a major achievement in the field. In particular, they can observe the position of an incoming nucleotides for the RNA synthesis in the catalytic core of the enzyme that has never been observed in any previous work. The manuscript is well written, describes perfectly the observations and therefore deserves to be published in Nature Communications.

We thank the reviewer for the positive evaluation!

Different aspects should also considered before acceptance.

Major

Regarding the manuscript, my major criticism concerns the way the manuscript is written, in particular how the authors consider the works of Grimes et al. (DOI: 10.21203/rs.3.rs-4663080/v1) and Yang et al. (Nat Commun; PMID: 39375338), published on the same subject 2024 July 12th and October 7th respectively. It means that the summary of the paper is not true, i.e. “However, to date no structural data is available on the NiV RdRp complex” as well as different sentences in the main text. If the three manuscripts (this one and the two already published) had a similar timeframe, this could have been perfectly acceptable, but that is not the case.

We agree with the reviewer that the manuscript should appropriately reflect the state of the literature. During the preparation of our manuscript, two preprints were posted, both describing the structure of the apo NiV RdRp complex: Grimes et al. (2024) and Abraham and Fearn et al (2024) (<https://doi.org/10.1101/2024.05.29.596445>). Since none of them had been published as peer-reviewed articles by the time we submitted our work, we cited them and considered our work to be complementary and in approximately the same timeframe. The

manuscript by Yang et al. (PMID: 39375338) was published in the same week we submitted our manuscript to Nature Communications and after we posted our work as preprint on bioRxiv, and we were hence not aware of this paper during the preparation of our manuscript. In the revised manuscript, we now cite all three other papers as well as two additional papers on the apo NiV L-P complex that were published in December 2024 (PMID 39661676, 39627254). We have also revised our wording in the summary and main text to emphasize that the novelty of our works lies in the RNA-bound structure.

The paper proposed by the authors bringing much more details than the two others do (I fully appreciate the quality of the work done), so the authors should take full advantage of this situation. As an example, Yang et al. with their incomplete structure speculate vaguely on the role of D832 for the transcription, whereas in the present active structure, this residue appears central for the RNA synthesis process and so crucial for the whole activity of the complex. Several aspects of the manuscript should therefore be reviewed to take these facts into account (with proper referenecees).

We thank the reviewer for this suggestion and have revised our manuscript to take into account the existing literature. In particular, we now discuss the results by Yang et al and others regarding D832 in the light of our structure.

The authors use the abbreviation “CAP” to refer solely to the GDP polyribonucleotidyltransferase domain (please add GDP line 59) that initiates the capping process at the 5’ of the viral mRNA. Considering the biochemical aspect, especially the cap being the guanosyl moiety (added by the PRNTase activity) as well as the methyl in position 7 of the ring (added independently from the first activity by the methylase domain), I think the use of this abbreviation can be confusing. Furthermore, considering also that the RdRp of segmented NSV such as Orthomyxoviridae and/or Bunyaviridae having a Cap-binding domain, this could so be a second source of discrepancy with the use of such trivial/easy-to-use abbreviation.

We agree with the reviewer that the designation “CAP” does not fully reflect the biochemical function of this domain, as it only carries out one step of the capping reaction. We have thus changed this and now refer to this domain as “PRNTase”.

Minor

- Please, check the names/abbreviations of the journals in the references: Example for ref 9 & 15 with PNAS with or without “.”. Idem ref 5 & 30 with J Virol with or without “.”. Please homogenize all the references.

We have corrected and homogenized the references.

- all Latin-sounding virus family names (Paramixoviridae, Filoviridae, ...) might be in italics

We have adopted this.

- homogenize “%” and “°C”. X% and X % (with space between numbers and %) as well as X°C and X °C can be found!

We have harmonized this.

- The Greek letter “gamma” used looks more a “y” than a real “γ”. Also doubts for the “β”

We have corrected this.

Response to the reviewers

Responses are in italique.

Changes are marked in red in the manuscript file.

Reviewer #1 (Remarks to the Author):

The manuscript has been significantly improved after revision. However, I do have two extra minors before the manuscript is appropriate for publication.

We thank the reviewer for the carefull evaluation.

1. Line480-481, the authors provided the supplier for GMPPNP, but the suppliers of other nucleotides, especially fluorescein-labelled CTP, are still not listed. In addition, the concentration of fluorescently-labeled CTP was reported as 150 mM, while the concentrations of other nucleotides were stated to be 1 mM. Authors need to double check the concentrations of all these nucleotides utilized in the manuscript.

We have now included the supplier and concentrations for all the nucleotides used.

2. Line 443-444, the authors state the peak fractions that eluted from the heparin column were identified by SDS/PAGE (Extended Data Figure 1b). The coming question is whether the sample used for mass spectrometry was also from the heparin column elution rather than from the size exclusion chromatography (line 91). The SDS-PAGE shows two Phosphoprotein bands (band2 band3 in Extended Data Figure 1b) were present in heparin column elution. Does the elution fraction of the SEC still have these two bands?

In fact, the mass spectrometry and SDS-PAGE shown in Extended Data Figure 1b correspond to the protein concentrated after SEC, which still contains the two bands. The heparin-purified sample showed degradation, but the SDS-PAGE for this step was not displayed. We have revised the text to improve clarity.

Reviewer #2 (Remarks to the Author):

My concerns have been addressed.